# Inflammatory Signatures of Maternal Obesity as Risk Factors for Neurodevelopmental Disorders: Role of Maternal Microbiota and Nutritional Intervention Strategies

**DOI:** 10.3390/nu14153150

**Published:** 2022-07-30

**Authors:** Francesca Cirulli, Roberta De Simone, Chiara Musillo, Maria Antonietta Ajmone-Cat, Alessandra Berry

**Affiliations:** 1Center for Behavioral Sciences and Mental Health, Istituto Superiore di Sanità, Viale Regina Elena 299, 00161 Rome, Italy; chiara.musillo@iss.it or; 2National Center for Drug Research and Evaluation, Istituto Superiore di Sanità, Viale Regina Elena 299, 00161 Rome, Italy; roberta.desimone@iss.it (R.D.S.); mariaantonietta.ajmone-cat@iss.it (M.A.A.-C.); 3PhD Program in Behavioral Neuroscience, Department of Psychology, Sapienza University of Rome, 00185 Rome, Italy

**Keywords:** maternal obesity, inflammation, microglia, gut microbiota, high-fat diet, oxidative stress, nutritional intervention strategies

## Abstract

Obesity is a main risk factor for the onset and the precipitation of many non-communicable diseases. This condition, which is associated with low-grade chronic systemic inflammation, is of main concern during pregnancy leading to very serious consequences for the new generations. In addition to the prominent role played by the adipose tissue, dysbiosis of the maternal gut may also sustain the obesity-related inflammatory milieu contributing to create an overall suboptimal intrauterine environment. Such a condition here generically defined as “inflamed womb” may hold long-term detrimental effects on fetal brain development, increasing the vulnerability to mental disorders. In this review, we will examine the hypothesis that maternal obesity-related gut dysbiosis and the associated inflammation might specifically target fetal brain microglia, the resident brain immune macrophages, altering neurodevelopmental trajectories in a sex-dependent fashion. We will also review some of the most promising nutritional strategies capable to prevent or counteract the effects of maternal obesity through the modulation of inflammation and oxidative stress or by targeting the maternal microbiota.

## 1. Introduction

The perinatal environment plays a pivotal role for the developing mammal and, as postulated by the Developmental Origin of Health and Disease (DOHaD) theory, negative experiences, such as a suboptimal intrauterine environment, may set the stage for later life vulnerability to diseases [1]. To this regard, malnourishment during pregnancy, either deriving from the lack or from the excess of nutrients, can be considered as a maternal metabolic stress that can negatively affect fetal programming increasing the risk for metabolic disorders and cardiovascular diseases in the offspring [2,3,4,5]. Interestingly, while the pioneering studies in this field of research, for historical reasons, were framed in the context of hunger and starvation nowadays we have to face the sequelae of malnourishment due to the global spread of obesity. In fact, globalization and urbanization have gradually led to the so-called nutrition transition namely a reduction of physical activity associated with the increase in the consumption of “junk food” (i.e., low-cost food easily accessible, ultra-processed, hypercaloric and poor in nutrient) [6,7]. Worth to mention, a number of long-term longitudinal studies have provided evidence for an association between maternal obesity and an increased risk to develop cognitive disabilities, autism spectrum disorders, attention deficit hyperactivity disorder, anxiety and depression in the offspring [6,8,9,10]. In this context, it is important to highlight that obesity and the consumption of “junk food” most often co-occur in humans making difficult to discriminate the effects of the nutritional aspects per se from those related to maternal metabolic profile in the offspring. Thus, it is possible to hypothesize that overall derangements in the prenatal metabolic environment hold similar effects as those observed as a result of maternal psychological stressors, profoundly affecting fetal development and future life liability to mental health [6,8,11,12].

Despite the increasing body of evidence linking maternal obesity or stress to offspring mental health problems, the mechanisms underpinning these effects remain poorly understood.

Inflammation has recently obtained widespread attention as a likely mediating mechanism for its involvement in the pathophysiology of both obesity and mental disorders [13] and for the emerging role of inflammatory mediators as regulators of brain homeostasis and plasticity [14,15,16]. As a general mechanism, the expansion of adipose tissue observed in obese subjects is associated with increased size and number of adipocytes leading to adipocyte hypoxia, infiltration of leukocytes, inflammation and insulin resistance [17,18,19]. Preclinical and clinical studies clearly indicate that these conditions, which set a chronic state of low-grade systemic inflammation, are magnified during gestation [20,21,22], a time when tightly balanced immune changes naturally occur in non-obese pregnant women [23]. This chronic low-grade systemic inflammation is also maintained by the adipose tissue that, acting as a metabolic and endocrine organ, greatly affects the physiology of pregnancy, as well as fetal development [24] providing an overall suboptimal intrauterine environment here generically defined as “inflamed womb” [17,25,26]. As an example, in physiological pregnancies, the placenta is fairly protected from peripheral inflammation, with macrophages of both the decidua and placental typically observed in an anti-inflammatory condition to allow the immune tolerance of the growing fetus. However, in the setting of maternal obesity, histopathological evidence of placental inflammation is described suggesting that such immune tolerance may be disrupted [20,27,28]. Moreover, Melekoglu and colleagues have provided evidence for increased levels of inflammatory cytokines in the amniotic fluid of women with a body mass index (BMI) ≥35 [29]. Notwithstanding the prominent role played by the adipose tissue, other players may also greatly contribute to create an inflammatory state: one above all is the maternal gut microbiota. This population of trillions of symbiotic bacteria participates in the metabolic, biochemical, and immunological balance of the host organism, hence playing a central role for human health [30]. The composition and function of this bacterial community is tightly modulated by the environment. To this regard, treatment with antibiotic, consumption of “junk food” on a regular base, obesity, and chronic stress are all factors that may trigger a condition of dysbiosis within the gut bacterial community, i.e., a dramatic loss of the homeostatic balance between the beneficial and potentially pathogenic bacteria that has been associated with a great variety of pathological conditions (metabolic, autoimmune, psychiatric, neurodegenerative) related to systemic inflammation (see [31] and references therein). A growing number of clinical evidence suggests that the quality and patterns of diets by affecting gut microbiota may modulate mood and behavior [32,33,34]. Changes in the inflammatory profile characterize healthy pregnancies—often associated to changes in the redox balance—and are aimed at promoting embryo implantation and parturition [23]; such changes are accompanied by time-specific adaptations of the gut microbiota composition. This tightly regulated process when deranged by metabolic stressors results in a pro-inflammatory condition [8,35,36,37].

However, how do we go from an “inflamed womb”, such as that characterizing maternal obesity, to long-term changes in brain function in the offspring? A number of preclinical studies is starting to answer this question, showing consistent findings indicating that maternal obesity during pregnancy might specifically affect brain development by targeting microglia, the resident brain immune macrophages. These cells are of unique origin and function compared to their peripheral counterpart and, besides immune defense, they play a pivotal role in brain development, homeostasis and plasticity; their homeostatic functions as well as their response to local or systemic challenges are sex-dimorphic and age-specific [38,39,40,41,42,43,44,45]. Alterations in microglial function in the offspring of obese dams appear to be long-lasting and associated to inflammation and changes in the architecture of specific brain areas overall resulting in decreased cognitive abilities [41]. Interestingly, a direct link between maternal microbiota and the development of embryonal microglia has been recently reported providing evidence for microglia derived from germ free (GF) mice to show an altered expression of those genes involved in the immune response and to be characterized by an overall immature or hypoactive immune state [46]. Although microglia might be only one of the many players, in this review we have taken into account also the hypothesis that maternal obesity-related dysbiosis and the associated inflammation might target this specific cell type altering neurodevelopmental trajectories in a sex dimorphic fashion. Moreover, we have reviewed some of the most promising nutritional strategies to prevent or counteract the effects of maternal obesity both in terms of their efficacy in modulating oxidative stress (OS) and inflammation, and their ability to target the maternal microbiota. Relevant literature to the main topic of this review has been retrieved by searching on PubMed.

## 2. Microglia as a Main Target of Maternal Obesity-Derived Inflammation

The chronic low-grade systemic inflammation characterizing maternal obesity is similar to that observed as a result of an early-life bacterial infection and suggests that common molecular and cellular mechanisms are activated by these different prenatal conditions [13,47,48]. The mechanisms mediating the propagation of maternal inflammation to the developing fetal brain are poorly elucidated, although the microenvironment generated by the placenta at the maternal/fetal interface is emerging as central in this phenomenon [13,49]. Altered ratios of innate and adaptive immune cell subsets and increased expression of placental pro-inflammatory and oxidative factors interfere with fetal development through several mechanisms, including maternal delivery of metabolites, nutrients, hormones, antibodies, and also cells, to the fetus [50]. Fetal brain microglia originate from a common erythromyeloid precursor in the extra-embryonic yolk sac, in which they are exposed to mother-derived signals since early developmental stages. Microglial progenitors colonize the developing brain as early as embryonic day 9 in rodents [51] and at the 4th–5th month and beyond in humans [52] where they persist and self-sustain throughout adulthood by local self-proliferation under steady state conditions. The estimated median lifespan of microglia is about 15 months in the mouse cortex and around 4 up to 20-years in humans. Their longevity and their remarkable capacity of long-lasting immune memory responses holds main implications for the quality and extent of the brain’s responses to new threats following a previous experience [53]. Microglia are key players in brain development, maturation, function and plasticity in healthy and disease conditions. By acquiring distinct phenotypes, with unique gene expression profiles, during the different phases of development they shape neuronal circuits and neuronal apoptosis to support and regulate neurogenesis and synapse plasticity [38,39]. Being the primary sensors of environmental changes in the central nervous system (CNS) they are key actors in mediating the effects of maternal obesity.

In the last decade, several preclinical studies of maternal obesity have shown evidence of how HFD-induced neuroinflammation might contribute to impaired social behavior in offspring of overfed dams. One of the first evidence demonstrating that the peripheral inflammation associated with maternal obesity is capable of programming microglial reactivity and inflammation within the offspring brain comes from the study of Bilbo and Tsang [40]. These authors found increased levels of the microglial activation marker cluster of differentiation molecule 11b (CD11b) within the hippocampus of newborn pups from HFD dams, as well as higher density of Ionized calcium-binding adaptor molecule 1 (Iba1) positive microglia in the hippocampus (CA1, CA3 and dentate gyrus) of male and female rat offspring at adulthood. They also found that a peripheral immune challenge at weaning and adulthood, despite the fact that the offspring themselves were maintained on a low-fat diet after weaning, induced a significant increase of the pro-inflammatory cytokine Inteleukin-1 beta (IL-1 beta) in the hippocampus of HFD offspring compared to controls. These alterations were accompanied by marked changes in anxiety-like behaviors and spatial learning. This evidence strongly supports that maternal obesity could prime offspring microglia towards a pro-inflammatory phenotype that is retained postnatally; although the underlying mechanisms have not been elucidated in this study, they are very likely to involve mechanisms of epigenetic reprogramming, as described in other phenomena of microglial immune memory [53].

The alterations in offspring’s behavior and CNS inflammation induced by maternal HFD have been shown to be sex, age and brain region specific. Exposure to a maternal HFD during gestation and or lactation in mice results in increased anxiety behavior in females and hyperactivity in male offspring while alterations in sociability are observed only in female offspring [54]. Interestingly, these behavioral changes are accompanied by increased mRNA expression levels of the pro-inflammatory genes *IL-1 beta*, *Tumor Necrosis Factor-alpha (TNF-alpha)* and *Iba1*, and by increased Iba1 staining intensity at 6 weeks of age in the amygdala—a region associated with anxiety and social behaviors—only in female offspring from HFD-fed dams. Switching maternal diet from HFD to control diet during lactation results in a significant decrease in *Iba1* transcripts in the brain suggesting that dietary intervention may be able to alleviate the impact of maternal HFD on offspring brain inflammation [54]. HFD was reported to induce distinct behavioral and inflammatory response profiles between adolescent and adult animals with also sex- and region- specific changes [55,56]. Adolescent animals perinatally exposed to maternal HFD, showed a decreased anxiety behavior associated with increased levels of glucocorticoid receptor transcripts in the hippocampus, but not in the amygdala, and by a dysregulation of inflammatory gene expression in both hippocampus and amygdala. Male and female adult rats showed instead increased anxiety behavior, concomitantly to a selective alteration in the expression of corticosteroid receptors in the amygdala in both sexes, and by higher levels of *nuclear factor kappa-light-chain-enhancer of activated B cells* (*NF-kB*) and *Interleukin-6* (*IL-6*) transcripts, two important inflammatory genes downstream to the glucocorticoid signaling, only in females.Winther and colleagues [57] found that the anxiogenic phenotype of the adult offspring born to HFD dams was correlated to an increase of the hippocampal mRNA levels of the pro-inflammatory cytokine *TNF-alpha* and of *monocyte chemoattractant protein-1* (*mcp-1*), an important chemokine that regulates microglial recruitment and activation. In addition, maternal HFD increased the offspring’s levels of hippocampal *corticosteroid releasing hormone receptor 2* (*crhr2*) and *kynurenine mono oxygenase* (*kmo*), whereas *kynurenine aminotransferase I* (*kat1*) mRNA levels were decreased. These observations strongly support that neuroinflammatory and stress-axis pathways may contribute to anxiogenic effects of maternal HFD [57].

More recently, Wijenayake and co-workers investigated the susceptibility of the adult offspring exposed to maternal HFD during perinatal life to an immune (lipopolysaccharide, LPS) and a stress (corticosterone, CORT) challenge or to their combination [58]. They observed that in response to CORT alone, male HFD offspring were more affected than females, showing increased levels of anti-inflammatory transcripts in the hippocampus and amygdala, whereas in response to LPS alone, female HFD offspring showed, in the same brain regions, increased levels of pro-inflammatory transcripts. Furthermore, in response to a CORT and LPS combination male HFD offspring showed greater pro-inflammatory transcriptional response in the cerebral cortex while female HFD offspring exhibited increased anti-inflammatory gene expression in the amygdala and cortex [58].

In another recent study, Bordeleau and colleagues found sex-dimorphic ultrastructural changes of microglia in maternal HFD-exposed offspring in the dorsal CA1 region of the hippocampus. They reported the presence of morphological microglial alterations and a reduction in microglia-associated pockets of extracellular space that may indicate diminished capacities to remove extracellular debris. While these changes were observed in both sexes, male offspring also showed increased microglia-astrocyte interactions as well as a reduction of mRNA expression of the inflammatory-regulating mediators *NF-kB* and *Transforming growth factor-beta* (*TGF-beta*), and the homeostatic microglial receptors *Transmembrane Protein 119* (*Tmem119*), *Triggering receptor expressed on myeloid cells 2* (*TREM2*) and *CX3C chemokine receptor 1* (*CX3CR1*), all key factors in the synaptic remodeling and inflammatory response [42].

Interestingly, a few studies in rodent models explored the combined effect of prenatal exposure to LPS with pre- and postnatal HFD on the immune and behavioral profile of the offspring [59,60,61]. Repeated LPS-stimulation of HFD dams during pregnancy affected the inflammatory profile in the offspring’s brain in a different way compared to a post-natal LPS exposure of maternal HFD offspring. The combined prenatal exposure (maternal HFD+LPS) lowered the IL-6/IL-10 ratio in the amygdala, hippocampus and prefrontal cortex, and reduced anxiety-like behaviors and short-term memory impairment at adolescence suggesting an unexpected protective effect of LPS against maternal HFD. Although in these studies microglial activation markers have not been evaluated, is reasonable that these cells play a key role in the protection induced by LPS, likely through the well-described mechanism of tolerance/sensitization, in which repeated exposure to an activating stimulus reduces macrophages/microglia responses to subsequent stimuli of similar or different nature and polarize them to anti-inflammatory functions [53,62]. These studies unveil poorly explored compensatory mechanisms of adaptation in the offspring brain.

The inflammatory response induced in HFD-fed dams and their pups by an excessive intake of dietary fats during pregnancy also affects myelination within the offspring’s cortex. Reduced levels of myelin basic protein (MBP), a mature oligodendrocyte marker, were revealed in the medial cortex of male offspring [63]. These changes were accompanied by a dysregulated brain iron metabolism and neurocognitive behavioral alterations, suggesting that the neuroinflammatory response induced in utero by maternal HFD may be correlated with a dysregulation of iron homeostasis and a reduction in myelination in male offspring [63].

Myelination alterations were also found within the corpus callosum—the region containing the most of the myelinated fiber projections within the brain—of HFD-fed dams’ offspring at adolescent stage [64]. Specifically, male adolescent offspring showed a reduction in the area of cytosolic myelin channels and of myelination-associated transcripts levels in the hippocampus, the projecting region of the corpus callosum. In addition, in both sexes, maternal HFD-exposed microglia exhibited increased numbers of contacts with synapses within the corpus callosum and reduced numbers of mature tertiary lysosomes suggesting an impaired phagolysosomal pathway [64]. Associated to these alterations, both male and female maternal HFD-adolescent offspring presented loss of social memory and sensorimotor gating deficits [64].

Collectively, these findings highlight how dysfunctional microglia and their crosstalk with neighboring glia may contribute to the adverse neurodevelopmental outcomes resulting from maternal HFD. Several other preclinical studies report how maternal HFD affects also hypothalamic inflammation, and microglial reactivity [65,66], as well as astrocyte proliferation [67]. It was shown that the proportion of proliferating astrocytes was significantly higher in the arcuate nucleus (ARC) and the supraoptic nucleus (SON) of the hypothalamus of maternal HFD offspring compared to control offspring from normal weight pregnancies. In addition, cultured fetal hypothalamic astrocytes proliferated significantly in response to IL-6, one of the cytokines significantly elevated in fetuses of obese dams, via the JAK/STAT3 signaling pathway [67] indicating that pathological exposure of developing hypothalamic astrocytes to cytokines would alter their development with repercussions for the developing brain.

The mechanism linking maternal obesity to long-term changes in brain function may depend upon complex interactions with the gut microbiome as microglia appear to be uniquely responsive to signals from the gut microbiome [30,42,46,68,69,70].

## 3. Obesity, Maternal Microbiota Dysbiosis and Its Sequelae on the Neurodevelopment of the Offspring

### 3.1. Role of the Microbiota in Human Health: The Forgotten Organ

The gut is the main site of interaction between the host immune system, commensals as well as pathogenic microbes in both mother and offspring. In fact, the gut microbiota is made up of a variety of microorganisms—mainly bacteria—playing a pivotal role in human health, so much so that it has been defined as the “forgotten organ” [71]. It is involved in the maintenance of the immunologic, hormonal, and metabolic homeostasis of the host and provides protection against pathogens; during the digestive process, it supports the extraction of energy and nutrients from foods serving as a source of metabolites, vitamins and essential nutrients. It synthesizes and releases neurotransmitters and neuromodulators, such as short-chain fatty acids (SCFAs), biogenic amines and other amino acid-derived metabolites playing a pivotal role in brain development, function and plasticity via a bidirectional exchange of signals between the gut and the brain occurring through the microbiota-gut-brain axis [30].

Physiological changes in the microbiota are observed throughout life, with life extremes (i.e., birth and senescence) being characterized by overt differences from the typical adult gut microbiota in terms of diversity, as well as in the representation of specific taxa [30]. Significant alterations in the microbiota composition are found in response to diet, BMI and stressful life events (among many conditions), greatly compromising its proper functioning leading to increased vulnerability to the onset of a variety of pathological conditions [59]. Alterations in the gut microbiota and inflammation may involve a bidirectional connection. In fact, while dysbiosis may promote the onset of pathological conditions such as metabolic syndrome, type 2 diabetes and inflammatory bowel disease (IBD), on the other hand, changes in microbiota composition may result from these same pathological states (see [32] and references therein). A similar bidirectional effect may apply also to neurodevelopmental and psychiatric disorders characterized by neuroinflammation through the microbiota-gut-brain axis [30,68,69,70] Changes in microbiota composition have been suggested to contribute to the endocrine, neurochemical and inflammatory alterations underlying obesity and the often-associated psychiatric disorders, a role that might be already at play during fetal life [33,34,65,72].

### 3.2. Microbiota Dysbiosis in Maternal Obesity

Clinical and epidemiological studies are consistently showing that the rise in the prevalence of obesity and metabolic diseases among women of the reproductive age has been paralleled by an increase in the occurrence of neurodevelopmental disorders in the offspring. According to the DOHaD theory, it is possible to hypothesize a direct relationship between metabolic derangements in the mother and neurodevelopmental outcomes in the offspring [73,74]. In this scenario, maternal microbiota is emerging as a key player providing a mechanistic link for the intergenerational negative effects observed as a result of metabolic maternal derangements. Specifically, and most importantly, maternal microbiota might contribute to the process of offspring brain development through different mechanisms, not mutually exclusive. One of the proposed mechanisms relies upon the phenomenon of the “leaky gut” a condition resulting in increased systemic and intrauterine inflammation due to obesity-related maternal dysbiosis. In particular, excessive BMI during pregnancy has been related to reduced alpha diversity and higher abundance of bacterial species such as *Staphylococcus aureus*, *Escherichia coli*, and in general of Proteobacteria, and lower counts of species such as *Bifidobacterium longum* and *Bacteroides fragilis* [75,76]. This abundance in Proteobacteria may lead to pro-inflammatory changes enhancing maternal gut permeability favoring the translocation of maternal intestinal bacteria to the fetus, eventually priming immune activation through the toll-like receptors (TLRs) [77,78]. Besides alive bacteria, many microbiota-derived microbial compounds such as LPS or flagellin may increase the production of pro-inflammatory cytokines (e.g., TNF-alpha, IL-6 and IL-1beta) activating the TLRs and affecting fetal development [78,79]. To this regard, studies performed in non-human primates and rodents have provided evidence that microbial maternal dysbiosis, deriving from HFD consumption, a model of maternal obesity, is associated with increased inflammation and is accompanied by elevated anxiety levels and reduced social behavior in the offspring [40,80].

As previously mentioned, bacterial metabolism is characterized by the production of SCFAs, macromolecules resulting from gut microbial anaerobic fermentation, that play a pivotal role in brain development, function, and plasticity. A growing body of evidence suggests that SCFAs play a pivotal role in linking maternal diet during pregnancy (and/or lactation) to inflammatory diseases (e.g., asthma) and obesity in offspring. In fact, being potent epigenetic modifiers and holding anti-obesity effects SCFAs have the potential to modulate host energy metabolism and the infant immune system through the interactions of leptin and insulin signaling in hypothalamic neurons, affecting food intake and energy expenditure (see [81,82,83] and references therein).

SCFAs concentrations increase significantly in maternal gut throughout pregnancy and recent preclinical studies have provided evidence for their beneficial effects on embryonal development acting as signaling molecules through G-protein receptors (GPR) GPR41 and GPR43 [84,85]. In particular, SCFAs have a major impact on fetal immune system development affecting cytokine synthesis and anti-inflammatory activities (see [78] and references therein).

Microglia are sensitive to intestinal microbiome changes and receive signals from the vagus nerve to regulate neuroimmune activity and function. Importantly, recent evidence suggests that microglia is developmentally regulated by the host microbiome and have been shown to dynamically respond to metabolic products from gut microbiota (SCFAs) in a sex-dependent manner [46,86]. Thion and collaborators [46] found that offspring born from GF mouse dams expressed changes in genes regulating LPS recognition and processing in utero and continued to exhibit sex-specific alterations in microglial-related gene expression postnatally [87]. Furthermore, altered microglial morphology and density, as well as attenuated inflammatory responses, were also observed right after birth, that is when microglia exhibit an activated phenotype [88]. It is important to note that the timeframe when the human gestational microbiota begins to stabilize resembling an adult composition—i.e., around three years of age—also overlaps with critical periods of CNS development, synaptic pruning, and neural remodeling. These observations support the hypothesis that a complex commensal microbiota ecosystem and their metabolites are integral to the early programming of key host physiological systems [87].

A further—though debated—mechanism through which changes in microbiota might contribute to modulate fetal brain development refers to the hypothesis of an intrauterine origin of fetal microbiota [77,78]. Given the importance of the microbiota-gut-brain axis for (neuro)development, gastrointestinal tract colonization represents indeed a milestone [89]. Although for about a century it has been hypothesized that the fetus is sterile, and that vaginal delivery is the first opportunity for large-scale bacterial colonization, recent studies are challenging the dogma of the “sterile womb” providing evidence for microbes to colonize the amniotic fluid, the umbilical blood cord and the placenta. In particular, the presence of an established microbiota in healthy and term infants supports the hypothesis that microbial colonization, occurring before birth, may play a role in the physiological development of the fetus [76,77]. See Figure 1.

## 4. Nutritional Strategies to Counteract Maternal Obesity

The sequelae of maternal obesity might become manifest early during development but also in the long run [90]. Excessive BMI before and during pregnancy has been associated with insulin resistance and gestational diabetes and is more likely to result in life-threatening conditions (both for the mother and the offspring) such as pre-eclampsia [91]. Moreover, it has been related to perinatal morbidity and to an overall increased chance for the offspring to develop metabolic, cardiovascular, and mental health disorders throughout life [8,14,92]. To this regard, an interesting meta-analysis performed by Thangaratinam and co-workers suggests that dietary interventions are the most effective way to contrast maternal obesity. Based on 34 different randomized trials, the mean reduction in maternal gestational weight gain due to the dietary intervention was −3.84 kg vs.−1.42 kg obtained through interventions such as moderate physical exercise [93]. Notwithstanding the great efforts of clinical practitioners to inform the general population about the risks related to excessive BMI, particularly during pregnancy, and the recommendation to reduce or contain gestational weight gain, compliance to diet regimens and to moderate physical exercise during pregnancy is very low and women are likely to maintain pre-pregnancy lifestyle until parturition (and even to gain weight during lactation) [94]. In this context safe and effective nutritional strategies, with limited side-effects, are becoming of utmost importance as rates of obesity continue to increase and the long-term negative effects on the offspring health is becoming more and more apparent [95]. Thus, addressing proper dietary supplementation might provide a broad-spectrum promising and feasible strategy to prevent or counteract the disruptive effects of the “inflamed womb”. In the following paragraphs, we will review nutritional strategies and dietary supplementation that have received recent attention and that appear to be promising based on the results obtained on preclinical studies and in clinical trials.

These intervention strategies include Long Chain Polyunsaturated Fatty Acids (LC-PUFA), Milk fat globule membrane (MFGM) and N-Acetyl-Cysteine (NAC) supplementation, all of which hold the potential to modulate inflammatory/oxidant processes and to target microbiota functions by multiple mechanisms, as described in the following paragraphs.

### 4.1. LC-PUFA Supplementation as Anti-Inflammatory and Protective Strategy

Obesity is associated with an altered profile of circulating fatty acids (FAs), including LC-PUFAs of the omega-3 and omega-6 series which is exacerbated by pregnancy-related adaptations. In both humans and animal models of maternal HFD, increased circulating levels of n-6 PUFAs, mainly arachidonic acid AA (AA, C20:4n-6) and linoleic acid (LA, C18:2n-6), and decreased levels of n-3 PUFAs, mainly docosahexaenoic acid (DHA, 22:6n-3) and eicosapentaenoic acid (EPA, C20:5n-3) were found. These alterations in maternal circulating levels are mirrored in breast milk and fetal circulation [96,97]. As the LC-PUFA supply to the fetus and the newborn relies mainly on their transfer from the maternal circulation across the placenta and from the breast milk, the dysregulation of maternal lipid metabolism profoundly affects the lipid profile of the fetus in different organs, including the brain [98,99].

In addition, the placental uptake and metabolism of essential FAs, particularly of DHA, is impaired by maternal obesity [96,97] and this occurs in a sex-specific manner, reflecting the differential expression, content, and activity of placental FAs transport and translocator proteins, as well as metabolizing enzymes [100,101]. Males’ offspring placentas showed reduced DHA-transfer capacity and the umbilical cord plasma had lower DHA-phospholipids than in females, which could lead to a lower DHA supply for proper fetal development in males under metabolic stress conditions [100].

The elevated n-6/n-3 PUFA ratio in the mother is associated with adverse pregnancy outcomes, shorter length of gestation and impaired fetal growth [102] being a key factor in the onset of low-grade, chronic inflammation associated with obesity in the mother-fetus dyad.

Long-chain fatty acids are indeed crucial regulators of the functional polarization of macrophages, through a multiplicity of mechanisms yet to be fully elucidated. These include the modulation of membrane fluidity—and hence of membrane-associated enzymes, receptors and signaling pathways—and the direct agonism to extracellular or intracellular receptors and transcription factors, such as G protein-coupled receptor 120 (GPR120), retinoid X receptors (RXR) and Peroxisome Proliferator-activated Receptor-gamma (PPAR-gamma), involved in the regulation of inflammation. AA and AA-derived eicosanoids act mainly, though not univocally, as pro-inflammatory mediators [103,104,105], while EPA, DHA and the DHA-derivatives docosanoids are recognized anti-inflammatory pro-resolving mediators (see [103,104,105] and references therein).

A recent study from Belcastro and colleagues found that placentas from women with pre-gestational obesity contain overall higher contents of PUFAs and a higher ratio of AA-derived isoprostanes vs. DHA-derived neuroprostanes, which are pro- and anti-inflammatory bioactive molecules, respectively [106]. Neuroprostanes were negatively correlated with inflammation markers, while isoprostanes were positively associated with maternal pre-gestational BMI [107], suggesting the importance of rebalancing PUFA levels in obese pregnant women as an anti-inflammatory/protective strategy for the progeny.

In the Fat-1 transgenic mouse model, capable of producing n-3 fatty acids from the n-6 type, Heerwagen and colleagues demonstrated that high maternal n-3/n-6 ratio prevents the consequences of maternal HFD-associated inflammation on weight gain, body and liver fat, and insulin resistance, and limits the development of adverse metabolic outcomes in adult offspring [108]. In line with these findings, DHA supplementation or higher fish consumption as a dietary omega-3 source in obese pregnant women decrease placental inflammation [109,110] further supporting that re-balancing n-3/n-6 LC-PUFA ratio represent a promising interventional approach to counteract the negative effects of maternal obesity and related inflammation.

The re-balancing of the n-3/n-6 LC-PUFA ratio could be crucial also for the proper fetal brain development. Notably, maternal obesity and maternal dietary n-3 PUFA deficiency leads to similar defects in prenatal brain development, immune status and behavioral impairments in animal models. Both maternal HFD and maternal low n-3 PUFA studies reported long-lasting changes in basal microglial reactivity and functions in the progeny, and higher susceptibility to subsequent inflammatory stimuli, as well as marked changes in anxiety and spatial learning behavioral patterns, as also mentioned in the previous section [40,42,111,112,113].

In mice, n-3 PUFA deficient maternal diet leads to increased n-6 PUFA levels in microglial membranes, while maternal n-3 PUFA dietary supply increases microglial DHA membrane content [113], demonstrating the strict correlation between maternal n-3 PUFA intake and microglial lipid profile in the developing brain, with likely repercussion for microglial reactivity and brain development.

Indeed, in activated brain microglia as in peripheral macrophages, DHA inhibits TLR inflammatory signaling, enhance fatty acid metabolism-related genes, reduces the synthesis of pro-inflammatory and anti-neurogenic molecules, and favors the shift to a phenotype that promotes the survival and differentiation of neural precursor cells [114,115,116,117,118,119,120,121]. Madore and colleagues recently reported that DHA is also crucial in maintaining the homeostatic molecular signature of microglia in physiological conditions and in tuning microglia-mediated phagocytosis of synaptic elements in the rodent developing hippocampus. In particular, a maternal n-3 PUFA deficient diet dysregulated microglial homeostasis in the progeny, increasing microglial immune and inflammatory pathways, and enhancing the phagocytosis of synaptic elements leading to their excessive removal [122].

Besides their immunomodulatory properties, n-6 (particularly AA) and n-3 (particularly DHA) PUFAs play essential roles for neural and visual development, as structural and functional components of neural membranes [123]. Incorporated into membrane phospholipids in uniquely high levels in the CNS, with the highest demand during the third trimester of pregnancy and the first two years of postnatal life, they promote neural and glial cell growth and differentiation [124,125,126]. For example, DHA was shown to promote oligodendrocyte progenitor cell maturation through its PPAR-gamma agonistic activity and the activation of extracellular signal-regulated-kinase (ERK)-1/2 [127]. Consistently, Leyrolle et al., 2021 recently demonstrated that maternal dietary n-3 PUFA deficiency impairs oligodendrocyte maturation and myelination during the postnatal phase, with long-term consequences on white matter structure, brain functional connectivity, mood, and cognition in adult mice. Other studies, in experimental models and humans, illustrated that LC-PUFAs are essential to astrogliogenesis, neurogenesis, synaptic plasticity, and neuronal wiring [128,129,130,131,132,133,134]. Collectively, these studies highlight the importance of an adequate dietary intake of LC-PUFAs, especially in highly sensitive windows of development since conception.

A recent study in mice fed a HFD suggested that high-DHA tuna oil significantly ameliorated obesity and metabolic dysfunctions in offspring by the regulation of taxonomic compositions of gut microbiota and their functional profiling, as the intestinal arginine and proline metabolism was restored by DHA supplementation [134]. Accumulating evidence from case reports and from animal studies suggests that DHA and other n-3 PUFAs can exert positive effects on gut microbiome signatures and composition [135,136,137]. More recently, it was shown that DHA supplementation repaired the lipids metabolism shifts in a mouse model of antibiotic-mediated gut dysbiosis [138]; similarly DHA significantly improved dyslipidemia and anxiety-like behaviors induced by long-term azithromycin (AZI) treatment in mice [139].

In the last decade several prenatal and postnatal clinical trials of n-3 PUFA supplementation have been conducted varying in sample size, dose and composition of supplements, timing and duration of intervention and demographic characteristics. The results regarding beneficial effects on offspring cognitive and/or metabolic outcomes remain controversial (see [140] and references therein). In addition, studies evaluating the role of n-3 PUFA supplementation specifically in the higher risk cohort of women with higher baseline metabolic dysregulation and with low n-3 status are limited to a pilot study from Monthé-Drèze and colleagues [141]. Supplementation with n-3 PUFA in women with obesity during pregnancy was associated with increased fetal fat-free mass accrual, improved fetal growth, and increased length of gestation. Larger adequately powered trials of n-3 supplementation or dietary intervention, starting before conception or early in pregnancy, specifically in women with excessive overweight should be conducted to confirm these findings.

### 4.2. Milk Fat Globule Membrane

The milk fat globule membrane (MFGM) is a unique, complex structure surrounding milk fat globule. It is essentially an oil droplet enclosed in plasma membrane that is secreted from the milk-producing cells in the epithelium of the mammary gland (lactocytes) and is physiologically delivered to the newborn through breastfeeding. It is a rich source of bioactive compounds including phospholipids, sphingolipids, gangliosides—and polar lipids in general—that hold a main functional role for the proper development of both the brain and the gut, two organs that communicate bidirectionally through the gut-brain axis system [142]. Moreover, MFGM has also the potential to synergize with probiotic bacteria—through direct interactions with proteins of the bacterial surface—overall increasing the survival and adhesion of probiotic bacteria during the gastrointestinal transit, improving mucosal immunity, and neurodevelopment and cognitive abilities in developing infants [143].

For the sake of clarity, in this paragraph, we will first review the beneficial effects of MFGM in postnatal development and adulthood and only successively we will hypothesize about the potential use of this compound in obese mothers during pregnancy to counteract the “inflamed womb” and its sequelae. Supplementation with dietary MFGM or with selected components thereof has shown beneficial effects on brain function, microbiome composition and immune system in several clinical and preclinical studies [142,144,145,146].

Breastfeeding appears to be positively related to cognitive and immune outcomes and to be overall associated with a decreased incidence of all-cause infection-related mortality in infants [142]. Recent preclinical evidence clearly suggests that MFGM-based interventions targeting the microbiota-gut-brain axis are able to buffer stress-induced dysfunctions of physiological processes and brain development [147,148]. Given the complex nature of this compound the pathways affected by MFGM are multiple and non-mutually exclusive and therefore quite a number of mechanisms are currently under investigation to clarify its positive effects on health outcomes during development and at adult age. Among all, MFGM has been shown to hold anti-inflammatory properties. Preclinical studies have provided evidence for laboratory rodents supplemented with MFGM to be characterized by a greater ability to counteract inflammation upon an immunogenic challenge. As an example, Snow and co-workers showed that MFGM-treated weanling mice displayed lower levels of inflammatory cytokines and reduced intestinal permeability upon LPS injection [149]; Huang and colleagues provided evidence for low-birth-weight mouse pups supplemented with MFGM during the suckling period to be characterized by reduced cytokines expression levels in the ileum (TNF-alpha, IL-6, IFN-gamma, IL-1beta) upon LPS treatment [150].

Recent evidence by Arnoldussen and co-workers has shown that MFGM was able to attenuate neuroinflammation in a mouse-model of HFD by reducing microglia activation resulting in better spatial memory performance and functional connectivity in the hippocampus, an effect possibly resulting from the modulation of phospholipid and metabolite composition in brain tissue or by increasing the level of gangliosides [151]. Besides their anti-inflammatory effects, gangliosides are known to be involved in neuronal transmission and are believed to support myelination, neuronal and synaptic plasticity during postnatal development. However, the rate of gangliosides accretion in the developing brain is highest in utero and in very early postnatal phases. Preclinical research has shown that complex milk lipids (also contained in MFGM) are transferred across the placenta and as such, an increased maternal intake during pregnancy has been associated with increases in fetal gangliosides levels [152,153]. Interestingly, a recent multicenter randomized controlled trial—the CLIMB (Complex Lipids In Mothers and Babies) study—was aimed to investigate the effects of supplementation of complex lipids in pregnancy, on maternal ganglioside status and subsequent cognitive outcomes in the offspring [154]; the authors report the absence of any adverse outcome but no effect of MFGM was reported on fetal growth, at least soon after birth, and for the parameters investigated in a population of healthy women [155]. Despite this inconclusive result, preclinical evidence rather suggests MFGM to be very promising to improve the health of the offspring. As for example, Yuan and co-workers report that supplementation of MFGM to obese rat dams during pregnancy and lactation improves neurodevelopment and cognitive function in male offspring. In particular, these authors reported that while maternal obesity induced insulin resistance and aberrant brain-derived neurotrophic factor (BDNF) signaling in the hippocampus of neonatal and adult offspring, these effects were counteracted by maternal MFGM administration [156]. Moreover, Li and colleagues provided evidence for the supplementation during pregnancy of MFGM to promote brown/beige adipocyte development and to prevent obesity in male offspring born from HFD-dams rats [157]. These same authors showed that MFGM may act through the modulation of gut microbiota to alleviate obesity-induced glucose metabolism disorders in peripheral tissues in rat dams [158]. Indeed, as mentioned above, MFGM is able to decrease gut permeability, which could attenuate gut-derived endotoxin translocation and the associated inflammatory responses supporting its use during pregnancies characterized by a general inflammatory condition such as those associated to obesity or hypercaloric unbalanced diets. Very interestingly, recent preclinical evidence shows that MFGM supplementation could reduce high-fat diet (HFD)-induced body weight gain and may control gut physiology and the amount of bacteria of metabolic interest [159]. Moreover, a clinical study showed that in overweight/obese adult subject, MFGM supplementation was able to reduce postprandial inflammation in a population characterized by a chronic inflammatory state due to obesity. In particular, MFGM lowered fasting plasma cholesterol, inflammatory markers as well as postprandial insulin response [160]. Thus, MFGM appears to be a safe and promising compound with the great potential to counteract the negative effect of maternal obesity promoting health outcomes of both the mother and the offspring.

### 4.3. The Antioxidant N-Acetyl-Cysteine as an Anti-Inflammatory and Protective Strategy in the Maternal-Fetal Crosstalk

OS and inflammation are main features of obesity and play a pivotal role in mediating the negative effects of this condition during pregnancy for both the mother and the offspring [8,92,161]. Indeed, a tight regulation of the redox balance and inflammatory processes, particularly during pregnancy, is crucial for a proper fetal development [14,162]. Moreover, OS has been reported to impair human placental amino acid uptake and increase Na+ permeability, directly affecting amino acid transporters [163].

To this regard, antioxidant compounds are emerging as a promising strategy for the treatment of diabetes and other metabolism-related pathologies [164]. Moreover, given the role played by OS and specific reactive oxygen species (ROS) in mediating fat accumulation, the reduction of OS through antioxidant compounds administration might mimic caloric restriction contributing to limit body weight gain [165,166,167,168]. Among the antioxidants, N-Acetyl-Cysteine (NAC) is receiving growing attention as a feasible pharmacological strategy in counteracting the detrimental effects related to maternal obesity [162,169]. NAC is the rate-limiting substrate in the biosynthesis of glutathione (GSH) and holds a great efficacy as a ROS scavenger [164]. We have previously shown that administration of NAC during pregnancy in a mouse model of maternal obesity resulted in improved glucose tolerance, in addition to a reduced activation of the HPA axis, when exposed to stress. Moreover, while HFD reduced antioxidant defenses (glutathione GSH levels) in the hypothalamus of male offspring, this effect was prevented by prenatal NAC administration. Since the hypothalamus plays a key role in both metabolic regulation and emotional behaviors, our data support a role for OS in the long-term effects of maternal obesity [162]. Interestingly, the above-mentioned results on NAC supplementation appear to mirror the protective effects observed in a genetically modified mouse model of reduced OS (p66Shc-/- mice) confirming the tight relationship existing among prenatal metabolic stress (maternal obesity), fetal programming and OS pathways [92,162,166,170]. NAC also holds anti-inflammatory capacity, being able to reduce the expression of pro-inflammatory mediators by suppressing the activation of the transcription factor NF-kB, whose regulation is redox sensitive [171,172,173] As such, NAC has been reported to reduce oxidative and inflammatory responses at the maternal-fetal interface and to improve placental efficiency in a number of clinical and preclinical studies. As an example, Williams and co-workers reported that NAC administration improved HFD-induced decidual vasculopathy in mouse placenta by reducing mRNA and immunostaining of IL-1beta and monocyte chemoattractant protein-1 and increasing *Vegf* m-RNA [174]. Paintlia and colleagues showed that NAC pretreatment was able to prevent LPS-induced preterm labor as well as inflammation and OS at the placental level and in the amniotic fluid and to inhibit LPS-induced up-regulation of phospholipids metabolism in placenta [175]. In the same rat model, these authors report that NAC was able to provide neuroprotection and to attenuate the degeneration of oligodendrocytes progenitor cells and white matter injury in the developing rat brain [176].

A number of severe pathological conditions of the newborns can be observed as a result of pre-eclampsia, a pregnancy-specific disorder characterized by new-onset hypertension and proteinuria that poses at great life-risk the mother and the newborn [177,178]. Pre-eclampsia shows a strong direct correlation with BMI and both inflammation and OS are involved in its etiopathogenesis. Although there is still no consensus in the literature on the best strategy for the prevention and the treatment of this disease, both anti-oxidant and anti-inflammatory approaches seem to be feasible [179]. To this regard, Motawei and colleagues observed attenuation of pre-eclampsia severity (improved liver and kidney function, decreased blood pressure, decreased proteinuria) and improved pregnancy outcomes among patients with pre-eclampsia who received NAC as they reported an increase in birth weight as well as a higher Apgar score [169].

As previously-mentioned one possible mechanism through which maternal obesity and maternal over-nutrition may trigger intrauterine inflammation—eventually affecting fetal brain development—is by inducing a condition of gut dysbiosis. To this regard, very interestingly, Luo and co-workers [180] showed that NAC supplementation in pigs during late gestation alleviated maternal-placental OS and inflammatory response, improved placental function, and altered fecal microbial communities. More in detail, NAC increased the relative abundance of fecal *Prevotella* that was positively related with the increased propionate and butyrate concentrations (two SCFAs able to reduce inflammation and OS). In addition, the relative abundances of *Clostridium cluster XIVa*, *Prevotella*, and *Roseburial/Eubacterium rectale* were related negatively to the gene expression levels of the NLRP3 and positively to that of Slc7a8 (that encodes for a transporter that mediates the uptake of amino acids in the placenta [181]. Overall evidence provided by Luo and co-workers indicates strong interactions between gut microbiota, placental NLRP3 inflammasome and nutrients delivery [180] and support the use of NAC as a promising compound to counteract the negative effect of maternal obesity. However, OS and inflammatory pathways need to be tightly regulated during pregnancy to ensure proper fetal development and a general consensus on the use NAC (and other antioxidants) during pregnancy in the clinical practice is still lacking [14,162].

## 5. Conclusions

A number of studies have clearly demonstrated an association between prenatal stress and neurodevelopmental disorders. Metabolic stressors, such as maternal obesity or malnutrition, due for example to the exposure to unbalanced diets rich in fats and sugars, can have similar long-term effects disrupting fetal programming. The role of the maternal gut microbiota in this context is still under study. However, being finely tuned throughout pregnancy and being very responsive to metabolic stressors, it could clearly represent an intriguing mediator capable to transmit the negative effects of maternal obesity/diet to the growing organism through multiple and non-exclusive pathways, leading to long-lasting inflammatory signatures.

In general, it is clear that the immune priming occurring in the fetal/newborn gut plays a pivotal role in brain development and in the health outcomes of the individual. This has been confirmed by a number of different experimental approaches ranging from GF animal (suggesting that microbiota plays a pivotal role in shaping both innate and adaptive immunity) to the manipulation of microbiota (either with antibiotic treatment or microbiota reconstitution providing evidence for the main role of the microbiota in immune homeostasis) [182]. Whether this priming occurs primarily in uterus or after birth is still a matter of debate however, it is also clear that the development of the immune system and the modulation of inflammatory responses both play a pivotal role in brain and metabolic health. This is also suggested by the growing number of studies reporting a comorbidity among autoimmune, psychiatric and metabolic disorders also in relation with changes in microbiota composition (see for example [183,184] and references therein).

In fetal brain, one important target of maternal microbiota is represented by microglia. Given the fundamental role played by this cell type on brain development, effective intervention strategies are needed in the future to break the vicious cycle that links maternal gut dysbiosis and inflammation to derangements in proper microglia development and function.

Overall, better understanding of the role of gut microbiota in maternal obesity might help us to promote healthy fetal brain and to prevent neurodevelopmental disorders.

## Figures and Tables

**Figure 1 nutrients-14-03150-f001:**
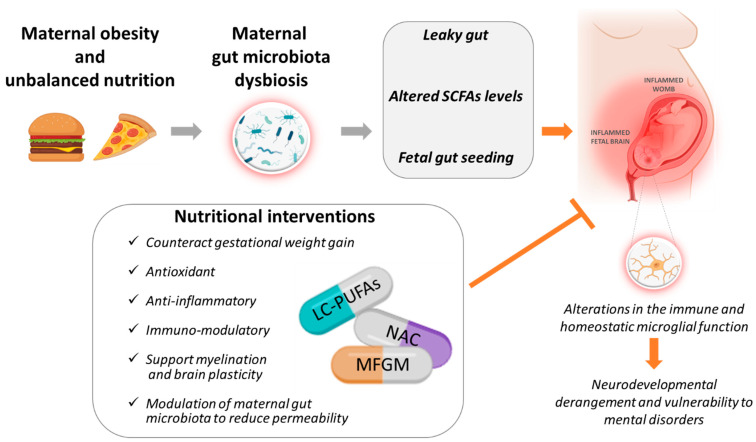
Obesogenic diets before and during pregnancy may trigger a dramatic loss of the homeostatic balance between the beneficial and potentially pathogenic bacteria in maternal gut leading to a condition of dysbiosis. This enhances gut inflammation that weakens the intestinal barrier (leaky gut) eventually resulting in the transplacental passage of bacteria and microbial compounds in the womb providing an immunogenic challenge to the fetus. SCFAs the main end-product of bacterial metabolism hold anti-inflammatory properties; they are tightly regulated during pregnancy to provide optimal fetal development. Obesity-related dysbiosis and inflammation may alter levels of SCFAs reaching the fetus. Finally, although still debated, the colonization of the fetal gut by maternally-derived bacteria (fetal gut seeding) in a condition of maternal dysbiosis might negatively affect the development of the gut-brain axis. The above-mentioned mechanisms may all contribute to provide a suboptimal intrauterine environment characterized by elevated systemic inflammation (inflamed womb) in turn affecting homeostasis in the developing fetal brain (inflamed brain) with microglia being a preferential target. Developing nutraceutical strategies, based on safe and feasible compounds (NAC, MFGM, LC-PUFA), aimed at counteracting gestational weight gain, reducing inflammation and oxidative stress is of paramount importance to support optimal brain development and to promote mental health throughout life. This image is original and has been created with BioRender.

## Data Availability

Not applicable.

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
