# Peer review of "Inflammatory Signatures of Maternal Obesity as Risk Factors for Neurodevelopmental Disorders: Role of Maternal Microbiota and Nutritional Intervention Strategies"

_nutrients, 2022, doi:10.3390/nu14153150_

Round 1

Reviewer 1 Report

The review article (#nutrients-1821661) by Francesca Cirulli et al. inflammatory signatures of maternal obesity as risk factors for neurodevelopmental disorders: role of maternal microbiota and nutritional intervention strategies.

The paper presents actual state of knowledge about the role of obesity as an inflammatory problem linked with prenatal programming. The article is well-written and gives a clear picture of the actual state of knowledge supported by the figure helping to understand the problem. It is very good review.

1. Abstract is consisted and informative

2. Introduction is well written, shortly presented the problem.

3. L49 – reference is needed

4. L 56- the term of “inflamed womb” should be replaced by other proper phrase. It is truth that obesity provides a low-grade systemic inflammation, but there is no evidence if uterus is inflamed and there are in the uterine muscle detected e.g. increased concentration of cytokines.

5. L 64- changes in the microbiota is one problem, and dysbiosis is separate problem

6. L 66 – reference should be added

7. L 68 – to whom it relates?

8. L73- the term is not adequate. Is there any evidence that uterine muscle is e.g. infiltrated by leucocytes? Increased volume but not by the pregnancy?

9. L85 – finish the sentence. The Style

10. L95- 119 – very well written part

11. L 121- the abbreviation should be explained

12. is maternal high fat diet equal to obesity?

12. L 132 – what is this?

13. L 144-145 – these symbols should be explained. What is this?

14. L 158, 158 – small letter

15. L 163- monocyte - small letter

16. L 187 -what is this?

17. L 216 – reference should be added

18. L 281- a good term

19. L 368 – not suitable term.

20. L 404- small letters

21. L 475 – AZI?

22. L 489-490- The milk fat globule is essentially an oil droplet enclosed in plasma membrane from the lactating cell. Add such information

23. L 522 – what does mean all these symbols?

24. L 637 – Metabolic stress may result from either lack or excess of nutrients and it has been recognized to occur in a number of living organisms. This term should be provided earlier in the introduction

Author Response

Reviewer#1

The review article (#nutrients-1821661) by Francesca Cirulli et al. inflammatory signatures of maternal obesity as risk factors for neurodevelopmental disorders: role of maternal microbiota and nutritional intervention strategies.

The paper presents actual state of knowledge about the role of obesity as an inflammatory problem linked with prenatal programming. The article is well-written and gives a clear picture of the actual state of knowledge supported by the figure helping to understand the problem. It is very good review.

  1. Abstract is consisted and informative

We thank the reviewer for the comment

  1. Introduction is well written, shortly presented the problem.

We thank the reviewer for the comment

  1. L49 – reference is needed

Done

  1. L 56- the term of “inflamed womb” should be replaced by other proper phrase. It is truth that obesity provides a low-grade systemic inflammation, but there is no evidence if uterus is inflamed and there are in the uterine muscle detected e.g. increased concentration of cytokines.

We thank the reviewer for the thorough comment. Indeed, we didn’t mean to refer to the organ itself (“uterus” that might have probably been the most appropriate term) but rather to the overall intrauterine environment that also include the placenta as well as the amniotic fluid. We are very sorry for the misunderstanding thus, to make the concept clearer to the readership and to avoid misunderstandings we have added the following:

Abstract: “Beyond the prominent role played by the adipose tissue, dysbiosis of the maternal gut may also sustain the obesity-related inflammatory milieu contributing to create an overall suboptimal intrauterine environment. Such a condition here generically defined as “inflamed womb” may hold long-term detrimental effects on fetal brain development, increasing the vulnerability to mental disorders”.

Introduction: “This chronic low-grade systemic inflammation is also maintained by the adipose tissue that, acting as a metabolic and endocrine organ, greatly affects the physiology of pregnancy, as well as fetal development [23] providing an overall suboptimal intrauterine environment here generically defined as “inflamed womb” [16,24,25]. As an example, in physiological pregnancies, the placenta is fairly protected from peripheral inflammation, with macrophages of both the decidua and placental typically observed in an anti-inflammatory condition to allow the immune tolerance of the growing fetus. However, histopathological evidence of placental inflammation are described in the literature suggesting that such immune quiescence may be disrupted in the setting of maternal obesity [26–28]. Moreover, Melekoglu and colleagues have provided evidence for increased levels of inflammatory cytokines in the amniotic fluid of women with a 35–40 BMI and >40 [29].”

  1. L 64- changes in the microbiota is one problem, and dysbiosis is separate problem
  2. L 66 – reference should be added

We thank this reviewer for pointing out this issue. The sentence has been rephrased as follows and appropriate references have been added: “The composition and function of this bacterial community is tightly modulated by the environment. To this regard, treatment with antibiotic, consumption of “junk food” on a regular base, obesity and chronic stress are all factors that may trigger a condition of dysbiosis within the gut bacterial community i.e. a dramatic loss of the homeostatic balance between the beneficial and potentially pathogenic bacteria that has been associated with a great variety of pathological conditions (metabolic, autoimmune, psychiatric, neurodegenerative) related to systemic inflammation (see [32] and references therein)”.

  1. L 68 – to whom it relates?

We thank the reviewer for pointing out this issue.

The sentence has been rephrased as follows to make it more clear to the readership: “A growing number of clinical evidence suggest that the quality and patterns of diets by affecting gut microbiota may modulate mood and behavior [33–35]"

  1. L73- the term is not adequate. Is there any evidence that uterine muscle is e.g. infiltrated by leucocytes? Increased volume but not by the pregnancy?

See above answer to Query 4

  1. L85 – finish the sentence. The Style

We thank the reviewer for the suggestion. The sentence has been rephrased as follows:

Interestingly, a direct link between maternal microbiota and the development of embryonal microglia has been recently reported providing evidence for microglia deriving from germ free mice to show an altered expression of those genes involved in the immune response and to be characterized by an overall immature or hypoactive immune state [45]

  1. L95- 119 – very well written part

We thank the reviewer for the comment

  1. L 121- the abbreviation should be explained

Done

  1. is maternal high fat diet equal to obesity?

We thank the reviewer for the thorough comment. The following sentence has been added to the Introduction: “In this context it is important to highlight that obesity and consumption of “junk food” most often co-occur in humans making difficult to discriminate the effects of the nutritional aspects per se from that related to maternal metabolic profile in the offspring.”

  1. L 132 – what is this?
  2. L 144-145 – these symbols should be explained. What is this?

We apologies for the typos, all Greek symbols have been replaced with the extended name of the letter

  1. L 158, – small letter

Done

  1. L 163- monocyte - small letter

Done

  1. L 187 -what is this?

We apologies for the typos, all Greek symbols have been replaced with the extended name of the letter

  1. L 216 – reference should be added

Done

  1. L 281- a good term

We thank the reviewer for the comment

  1. L 368 – not suitable term.

See answer to Query 4

  1. L 404- small letters

Done

  1. L 475 – AZI?

Acronym has been added

  1. L 489-490- The milk fat globule is essentially an oil droplet enclosed in plasma membrane from the lactating cell. Add such information

Done

  1. L 522 – what does mean all these symbols?

We apologies for the typos, all Greek symbols have been replaced with the extended name of the letter

  1. L 637 – Metabolic stress may result from either lack or excess of nutrients and it has been recognized to occur in a number of living organisms. This term should be provided earlier in the introduction.

We thank the reviewer for pointing out this issue. The term has now been better explained in the introduction.

See the following: “To this regard, malnourishment during pregnancy either deriving from the lack or from the excess of nutrients can be considered as maternal metabolic stress that can negatively affect fetal programming increasing the risk for metabolic disorders and cardiovascular diseases in the offspring [2–5]. Interestingly, while the pioneering studies in this field of research, for historical reasons, were framed in the context of hunger and starvation nowadays we have to face the sequelae of malnourishment due to the global spread of obesity. In fact, globalization and urbanization have gradually led to the so-called nutrition transition namely a reduction of physical activity associated with the increase in consumption of “junk food” (i.e. low-cost food easily accessible, ultra-processed, hypercaloric and poor in nutrient) [6,7].

Reviewer 2 Report

·      Abstract: The criteria adopted to perform the search strategy should be defined (e.g. Pubmed/Medline, Web of Science…etc).

·         Line 44-72: What about the evidence base linking maternal diet during pregnancy and/or lactation to obesity and its related inflammatory diseases (e.g., ND, asthma) in offspring? What is the role of gut microbiota? I would suggest authors referring to these reviews (Front Cell Neurosci. 2020; 14: 612705; J Neurosci Res. 2021 Jan;99(1):284-293; Nutrients. 2021 Oct 21;13(11):3702; Int J Mol Sci. 2020 Dec 16;21(24):9580).

·       In introduction, authors should mention that the gut is the main site of interaction between the host immune system and both commensal, as well as pathogenic microbes in both mother and offspring. How colonization by gut microbiota shapes the immune system? How the in vivo/vitro studies informed outcomes of the gut colonization?

·        The method section is missing. Do the paper reflect the existing literature about the topic, or the authors are showing an arbitrary choice of papers. Authors reviewed several studies but it is not clear how these studies were chosen for inclusion in the review. The section should include search terms, years conducted, study type (vivo/vitro), inclusion/exclusion criteria, databases …etc.

·         Line 49,54: Subjects/individuals- please clarify (e.g., mother, infants…etc).

·         Refs # 1,120 and144 are very old- please update.

Author Response

Reviewer#2

1) Abstract: The criteria adopted to perform the search strategy should be defined (e.g. Pubmed/Medline, Web of Science…etc).

See below answer to Query 2

2) The method section is missing. Do the paper reflect the existing literature about the topic, or the authors are showing an arbitrary choice of papers. Authors reviewed several studies but it is not clear how these studies were chosen for inclusion in the review. The section should include search terms, years conducted, study type (vivo/vitro), inclusion/exclusion criteria, databases …etc.

We thank the reviewer for the thorough comment. This manuscript is not meant to be a systematic but rather a narrative review based on a comprehensive, critical and objective analysis of the current most relevant evidence supporting the hypothesis that maternal obesity-related dysbiosis and the associated inflammation, might target fetal brain microglia altering neurodevelopmental trajectories in a sex-dependent fashion.

Although there is no specific requirement for a method section in a narrative review, we have taken into account your valid advice adding the following to the end of the Introduction “Relevant literature to the main topic of this review has been retrieved by searching on PubMed”.

3) Line 44-72: What about the evidence base linking maternal diet during pregnancy and/or lactation to obesity and its related inflammatory diseases (e.g., ND, asthma) in offspring?

What is the role of gut microbiota?

I would suggest authors referring to these reviews (Front Cell Neurosci. 2020; 14: 612705; J Neurosci Res. 2021 Jan;99(1):284-293; Nutrients. 2021 Oct 21;13(11):3702; Int J Mol Sci. 2020 Dec 16;21(24):9580).

We thank the reviewer for the thorough suggestion. We have incorporated this relevant literature in the text.

As for the Ref by Bordeleau and colleagues 2020 this was already mentioned in the Introduction.

Other suggested references have been added in the following sentences in paragraph 3.2 “A growing body of evidence suggest that SCFAs play a pivotal role in linking maternal diet during pregnancy (and/or lactation) to inflammatory diseases (e.g. asthma) and obesity in offspring. In fact, being potent epigenetic modifiers and holding anti-obesity effects SCFAs have the potential to modulate host energy metabolism and the infant immune system through the interactions of leptin and insulin signaling in hypothalamic neurons, affecting food intake and energy expenditure (see [82–84] and references therein)”.

4) In introduction, authors should mention that the gut is the main site of interaction between the host immune system and both commensal, as well as pathogenic microbes in both mother and offspring. How colonization by gut microbiota shapes the immune system? How the in vivo/vitro studies informed outcomes of the gut colonization?

We thank the reviewer for the thorough comment.

The sentence “The gut is the main site of interaction between the host immune system and both commensal, as well as pathogenic microbes in both mother and offspring” has been added at the beginning of paragraph 3.1

The following has been added to the Conclusions to stress the importance related to the colonization by gut microbiota in relation to immune system development also taking into account how the in vivo/vitro studies informed outcomes of the gut colonization:

“In general, it is clear that the immune priming occurring in the fetal/newborn gut plays a pivotal role in brain development and in the general health outcomes of the individual. This has been confirmed by a number of different experimental approaches ranging from GF animal (suggesting that microbiota plays a pivotal role in shaping both innate and adaptive immunity) to the manipulation of microbiota (either with antibiotic treatment or microbiota reconstitution providing evidence for the main role of the microbiota in immune homeostasis) [182]. Whether this priming occurs primarily in uterus or after birth is still a matter of debate however, it is also clear that the development of the immune system and the modulation of inflammatory responses play a pivotal role in brain and metabolic health. This is also suggested by the growing number of studies reporting a comorbidity among autoimmune, psychiatric and metabolic disorders also in relation with changes in microbiota composition (see for example [183,184] and references therein)”.

Moreover, paragraph 3.1 also generally contributes to the discussion in the relation to the topics suggested by this reviewer, see for example: “To this regard, studies performed in non-human primates and rodents have provided evidence that microbial maternal dysbiosis, deriving from HFD consumption, a model of maternal obesity, is associated with increased inflammation and is accompanied by elevated anxiety levels and reduced social behavior in the offspring [41,81].”…..”Thion and collaborators [47] found that mice born from germ free maternal mice expressed changes in genes regulating LPS recognition and processing in utero and continued to exhibit sex-specific alterations in microglial-related gene expression postnataly [85].”

5) Line 49,54: Subjects/individuals- please clarify (e.g., mother, infants…etc).

The sentences have been rephrased:

As a general mechanism, the expansion of adipose tissue observed in obese subjects is associated with increased size and number of adipocytes, leading to adipocyte hypoxia, infiltration of leukocytes, inflammation and insulin resistance [17–19]. Preclinical and clinical studies clearly indicate that these conditions, which set a chronic state of low-grade systemic inflammation, are magnified during gestation [20–22], a time when tightly balanced immune changes naturally occur in non-obese pregnant women [23]”.

6) Refs # 1,120 and 144 are very old- please update.

We thank the reviewer for pointing out this issue.

As for reference #1 this is extremely relevant and is the main reference for the Barker’s hypothesis as well as the successive Developmental Origin of Health and Disease (DOHaD).

Ref #120 (De La Presa Owens, S.; Innis, S.M. Docosahexaenoic and arachidonic acid prevent a decrease in dopaminergic and serotoninergic neurotransmitters in frontal cortex caused by a linoleic and α-linolenic acid deficient diet in formula-fed piglets. J. Nutr. 1999, 129, 2088–2093, doi:10.1093/jn/129.11.2088) and #144 (Hungund, B.L.; Morishima, H.O.; Gokhale, V.S.; Cooper, T.B. Placental transfer of (3H)-GM1 and its distribution to maternal and fetal tissues of the rat. Life Sci. 1993, 53, 113–119, doi:10.1016/0024-3205(93)90658-P. have been removed).